# Metabolomics-Guided Discovery of New Dimeric Xanthones from Co-Cultures of Mangrove Endophytic Fungi *Phomopsis asparagi* DHS-48 and *Phomopsis* sp. DHS-11

**DOI:** 10.3390/md22030102

**Published:** 2024-02-23

**Authors:** Jingwan Wu, Dandan Chen, Qing Li, Ting Feng, Jing Xu

**Affiliations:** Collaborative Innovation Center of Ecological Civilization, School of Chemistry and Chemical Engineering, Hainan University, Haikou 570228, China; 20081700110009@hainan.edu.cn (J.W.); 21220856000083@hainanu.edu.cn (D.C.); 22210710000026@hainan.edu.cn (Q.L.); 20081700210004@hainanu.edu.cn (T.F.)

**Keywords:** mangrove endophytic fungi, *Phomopsis asparagi*, *Phomopsis* sp., co-culture, metabolomics, molecular network, xanthone dimers

## Abstract

The co-culture strategy, which mimics natural ecology by constructing an artificial microbial community, is a useful tool for the activation of biosynthetic gene clusters (BGCs) to generate new metabolites, as well as to increase the yield of respective target metabolites. As part of our project aiming at the discovery of structurally novel and biologically active natural products from mangrove endophytic fungi, we selected the co-culture of a strain of *Phomopsis asparagi* DHS-48 with another *Phomopsis* genus fungus DHS-11, both endophyted in mangrove *Rhizophora mangle* considering the impart of the taxonomic criteria and ecological data. The competition interaction of the two strains was investigated through morphology observation and scanning electron microscopy (SEM), and it was found that the mycelia of the DHS-48 and DHS-11 compacted and tangled with each other with an interwoven pattern in the co-culture system. A new approach that integrates HPLC chromatogram, ^1^HNMR spectroscopy, UPLC-MS-PCA, and molecular networking enabled the targeted isolation of the induced metabolites, including three new dimeric xanthones phomoxanthones L-N (**1**–**3**), along with six known analogs (**4**–**9**). Their planar structures were elucidated by an analysis of their HRMS, MS/MS, and NMR spectroscopic data and the absolute configurations based on ECD calculations. These metabolites showed broad cytotoxic activity against the cancer cells assessed, of which compounds **7**–**9** displayed significant cytotoxicity towards human liver cells HepG-2 with IC_50_ values ranging from 4.83 μM to 12.06 μM. Compounds **1**–**6** exhibited weak immunosuppressive activity against the proliferation of ConA-induced (T-cell) and LPS-induced (B-cell) murine splenic lymphocytes. Therefore, combining co-cultivation with a metabolomics-guided strategy as a discovery tool will be implemented as a systematic strategy for the quick discovery of target bioactive compounds.

## 1. Introduction

Mangrove-derived microorganisms in intertidal microenvironments are tolerant to numerous stresses, including highly saline or brackish water, high solar irradiation, and tidal gradients. The special ecological niche induced certain buds being evolved to encode unique biosynthetic genes that have the potential to generate unique bioactive metabolites [1,2], which increasingly attracted the attention of both pharmaceutical and natural product chemists [3,4]. A therapeutic molecule from the mangrove environment is now available in the market and exemplified by salinosporamide A (Marizomib^®^), which is a fermentation product of mangrove-derived *Salinispora tropica* CNB-392 [5]. Salinosporamide A is a highly potent inhibitor of the 20S proteasome that was granted an orphan drugs designation by the European Medicines Agency (EMA) for the treatment of multiple myeloma [6]. It may be expected that further promising candidates obtained from mangrove-derived microorganisms will follow in the future.

Chemical investigations of mangrove-derived microorganisms, especially endophytic fungi, have shown a sharp increase in recent years, and more than 250 endophytic fungi strains produced 79% (1090) by the mangrove-derived fungi originated 1300 new compounds [7]. Nevertheless, genomic sequencing techniques began to reveal that most mangrove endophytic fungi possess significantly more biosynthetic gene clusters (BGCs) than the number of compounds they produce under standard laboratory culture conditions [8,9,10,11]. Several approaches have been developed to aid in the activation of these dormant BGCs by employing modern biological or chemical techniques, such as genomics, transcriptomics, proteomics, and metabolomics [12,13,14]. In addition to the techniques that demand prior knowledge of the genome sequences of the studied microorganisms, several genome sequence-independent tools have been developed [14]. One of these approaches adopted to enhance the expression of silent BGCs in fungal cultures is co-cultivation, which is based on the premise that two or more microorganisms growing within a confined environment respond to environmental cues, which trigger the activation of BGCs to produce often new, bioactive secondary metabolites that cannot be otherwise detected in the corresponding monocultures [15,16]. There are several insights to be gained from co-culture experiments with mangrove endophytic fungi, as the activated secondary metabolites encompass a broad range of structural diversity, such as alkaloids [17,18], isocoumarins [19], xanthones [20], phenols [21,22], and peptides [23,24]. These successes indicate that co-cultivation is a rational and effective approach to induce the production of new metabolites, as well as to increase the yields of respective target metabolites with pharmacological potential. However, studies examining the co-culturing the pairing endophytic fungi within the same mangrove host plant to mimic the co-exist occurring interactions in naturally ecological situation are limited and cannot confirm a conclusion.

Metabolomics is the comprehensive analysis of small molecule metabolites in a biological system to reflect a phenotype response to chemical and biological stimuli, providing insight into the biological functions [25]. Many spectroscopic (NMR, MS, MS–MS) and chromatographic (HPLC, GC, GC–MS, LC–MS, TLC, etc.) methods are widely used for metabolomic analysis [26]. The use of HPLC chromatograms followed by LC−MS-based principal component analysis (PCA) has been a useful tool to distinguish whether metabolomic changes engendered in the co-culture and monoculture samples are based on their LC–MS profiles [27]. Nonetheless, reliable dereplication is often very challenging when solely using MS-1 data and available databases, especially at *m*/*z* values <600 [28]. ^1^H NMR-based approaches have shown to be very effective in discriminating between alternative candidate structures in the dereplication process. It can provide comprehensive characteristic fingerprints since the co-cultivation can greatly affect the fingerprints of the extracts, based on structural features of specific chemotypes that can be easily determined from inspection of the chemical shift in the ^1^H NMR spectrum [29]. Meanwhile, molecular networking is the main analysis tool to create structured networking based on the observation that structurally related molecules share similar MS/MS fragmentation patterns. It has emerged as an effective technique for dereplication and offers information about the chemical diversity in selected extracts before isolation [30,31]. Particularly, it is available on the Global Natural Products Social Molecular Networking “http://gnps.ucsd.edu (accessed on 4 February 2024)” open-access web-based platform [32]. Indeed, a variety of studies have been conducted by employing metabolomics analyses coupled with molecular networking to screen the chemical diversity of fungal extracts and rapidly target the isolation of novel natural products from complex mixtures in co-cultivation [27,33,34,35].

In continuing the search for structurally novel and biologically active natural products from mangrove endophytic fungi [36,37,38,39,40,41,42,43,44,45], a strain of *Phomopsis asparagi* DHS-48 isolated from the fresh root of *Rhizophora mangle* was found to be particularly productive with regard to the accumulation of a series of immunosuppressive chromones and cytochalasins [46,47,48]. With the goal of tapping the metabolic potential of this titled fungal strain, we initiated the co-culture fermentation process of well-characterized DHS-48 with other fungal strains. Considering the impart of taxonomic criteria and ecological data (sharing an ecological niche), another *Phomopsis* genus fungus DHS-11 endophyted in *R. mangle* was selected for pairing, which is known as a producer of isocoumarin and pyrone derivatives [49]. The modifications of the morphological features of mycelia by the co-culture on a small-scale on solid potato dextrose agar medium (PDA) plates were investigated through scanning electron microscopy (SEM). The EtOAc extraction of the upscaled fermentation of the co-culture on a rice solid-substrate medium were subjected to metabolomics analysis of the mono- and co-cultures to observe specific induction of unexpressed pathways, such as HPLC chromatogram, ^1^HNMR spectroscopy, UPLC-MS-PCA, and molecular networking. A follow up metabolomics-guided isolation (Figure 1) led to the discovery of three new dimeric xanthones phomoxanthones L-N (**1**–**3**), along with six known analogs (**4**–**9**). Herein, we report on the co-culture morphological features and in-depth metabolomics analyses, followed by the isolation and structure elucidation of the induced dimeric xanthones (**1**–**9**) (Figure 1), as well as their cytotoxic and immunosuppressive activities.

## 2. Results and Discussions

### 2.1. Morphology of Co-Culture Systems

Co-cultivation can induce morphological changes and colony growth, leading to an interaction with the competing colony that permits the morphology of the species involved in the interactions to be monitored with the unaided eye [50]. Solid media was preferred over liquid cultures for the screening of induction phenomena in fungal co-cultures [51]. On the PDA medium, the *Phomopsis asparagi* DHS-48 colony appeared yellow–brown in the middle, but a concentric annular circle appeared on the reverse side of the medium and reached a diameter of 60 mm on day 7 (Figure 2A). When *Phomopsis* sp. DHS-11 was inoculated on the PDA medium at 28 °C, initially, it was cottony and white, then turned to yellowish, pale filaments around the agar plug. The fungal colony grew to 90 mm in diameter and covered the whole medium on day 7 (Figure 2B). When the DHS-48 and DHS-11 strains were co-cultured together on the PDA medium for 7 days, a dark black precipitate was observed in the confrontation fronts of the two fungal colonies in the middle, morphologically different from the axenic cultures (Figure 2C). It was revealed that the DHS-48 strain could coexist well with DHS-11, and the growth rate of the DHS-11 was higher than that of the DHS-48, with a larger DHS-11 colony on the PDA medium. The SEM provided keen insight on the mycelial microstructure, as well as that of the mat without considering the composition; however, it was able to determine the hyphae and porosity dimensions [52]. The SEM observation showed that the monocultures of DHS-48 had cylindrical to obclavate hyphae with slight surface wrinkles with a diameter of 31.87 ± 0.88 μm (Figure 2D). The diameter of the DHS-11 mycelia was about 11.78 ± 1.08 μm and was similar microscopically to DHS-48 (Figure 2E). The mycelia of the DHS-48 and DHS-11 compacted and tangled with each other with an interwoven pattern in the co-culture system (Figure 2F,G). The distinguishable competitive interactions between DHS-48 and DHS-11 denote significant metabolic induction phenomena are likely to occur, indicating the exploitation potential of stimulated new secondary metabolites that are not produced under standard monoculture conditions.

### 2.2. Metabolomics Analysis of Large-Scale Fermentation of Co-Cultures

An HPLC chromatogram with UV detection (HPLC-UV) of the crude EtOAc extract of a 30-day solid rice medium whole co-culture of DHS-48 and DHS-11 showed unexpected and significantly suppressed peaks and newly induced compounds **1**–**6** (Figure 3A), which were absent in the monocultures (Figure 3B,C). In addition, productions of the known metabolites **7**–**9** were substantially upregulated compared with their respective monocultures. Previous co-cultivation studies have provided evidence for the induction and suppression of biosynthetic pathways [53]. It is interesting to note that co-cultivation involving the suppression of BGCs or upregulation of known, functional metabolites mediates the competitive interaction by the other. The common characteristic UV absorption maxima around 203, 225, and 347 nm (UV data of compound **7**) showed a coincidence to related literature values [54], indicating that **1**–**9** should be dimeric xantone derivatives. Continuously, these differences were also supported by the fact that the ^1^H NMR metabolic profile (Appendix A) of the EtOAc extracts showed several significant enhanced hydrogen resonances at 6.5–7.7 ppm (aromatic protons), 3.5–4.5 ppm (alkoxy protons), and 2.5–2.75 ppm (methine/methylene protons) compared with the control groups.

A combination of ultra-high-performance liquid chromatography (UHPLC) and high-resolution mass spectrometry (HRMS) was employed to analyze the metabolomes. An unsupervised principal component analysis (PCA) was conducted to compare the metabolic features and identify statistically significant differences. All the co-cultures and monocultures had five independent biological replicates, ensuring the reliability of the results [55]. The examination of the scores plot revealed clustering of the samples in three groups, indicating that the co-culture fingerprints did not overlap with the two corresponding monoculture clusters (Appendix A). The separation meant that the datasets contained information that allowed for the discrimination of the chemical composition of the co-cultures from that of the monocultures, implying that microbial interactions modulated the biosynthetic pathways for the production of secondary metabolites.

In order to further investigate the metabolome and to obtain more information on the chemical diversity induced by co-culture, UPLC-ESI-MS/MS-based molecular networking was generated with the crude extracts of the fungal co-culture and the corresponding monocultures through the online Global Natural Products Social Molecular Networking (GNPS) platform. The generated molecular network (MN) (Figure 4A) consisted of 936 nodes in total, out of which 205 nodes were organized into 33 molecular clusters containing at least 2 nodes. As expected, co-cultivation significantly increased the chemical space of the fungi, indicated by the increased size of several molecular clusters with co-culture-induced derivatives (red-only nodes) of compounds (green or yellow nodes) produced in the axenic monocultures, such as clusters 1, 4, 6, 7, 12–14, 25, 27, and 30. Moreover, we observed seven clusters (8, 9, 15, 16, 24, 26, and 31) harboring only red nodes that were exclusively induced in the co-cultures. Unfortunately, no ion belonging to the latter clusters match to known compounds in the reference databases or could be purified in sufficient quantity to allow chemical identification. It is worth mentioning that the number of nodes in the network did not correspond exactly to the number of metabolites, as different adducts or charges of the same compounds could generate different nodes [56,57]. Additionally, it should be noted that a definite identification of any known compounds by MS/MS alone is not possible, as it is not possible to deduct the absolute configuration, and the potential presence of isomers cannot ultimately be ruled out [56]. Subsequent isolation efforts would be deemed worthwhile when the analysis of the GNPS clusters in question suggests the presence of potentially unknown compounds.

An in-depth analysis of the MN of the crude extract of the co-culture extracts and respective monocultures (Figure 4B) allowed us to track the distribution of nodes according to xantone dimers down to cluster 1 and two singletons. The biggest cluster 1, of which 92 nodes comprising several *m*/*z* values were found in a mixed origin of both co-culture and monocultures, was annotated to a xantone dimer class of compounds. The first node *m*/*z* 667.2021 [M+H]^+^ was determined to be the most abundant constituent dicerandrol A (**7**), previously described as a secondary metabolite isolated from the monoculture of DHS-48 [58], and dramatically up-regulated the amounts in the co-culture. It was purified and identified with the help of the MS/MS cleavage pattern (Appendix A), as well as a comparison with previously reported NMR data. The molecular network quickly allowed us to establish the structural relationships of dicerandrol A and two additional nodes at *m*/*z* 583.1811 [M+H]^+^ (compounds **5**/**6**) and *m*/*z* 709.2129 [M+H]^+^ (compound **9**) within this cluster. They were purified and identified as the known phomopsis-H76 A (**5**), diaporthochromone B (**6**), and 12-O-deacetyl-phomoxanthone A (**9**) (Appendix A). Node *m*/*z* at 751.2232 [M+H]^+^ (compound **8**) was found to cluster as an adjacent node with *m*/*z* 709.2129, which is 42 Da larger than that of *m*/*z* 709.2129 and suggested the presence of an extra acetyl group. A comparison of its fragment ions (Appendix A) with a manual search in the database confirmed the annotation of dicerandrol C (**8**). Another node detected at *m*/*z* 601.1916 [M+H]^+^ (compound **3**) directly linked to node *m*/*z* 583.1811 has a mass difference of 18 Da, suggesting high structural similarity and a putative H_2_O increment. The predicted molecular formulae C_30_H_32_O_13_ of this node returned no hits with a dimeric xantone scaffold, hence it is putatively new. We tentatively proposed its structure based on the high MS/MS spectral similarity score (>0.65) as implemented in the GNPS-MN platform (Appendix A). Follow-up targeted purification of *m*/*z* 601.1916 [M+H]^+^ (Appendix A) allowed the assignment of the structure as new phomoxanthone N (**3**), confirming the cluster annotation. Other putative annotation was based on manual dereplication, including deacetylphomoxanthone B (*m*/*z* 665.389 [M−H]^−^) and an isomer of curtisian C (*m*/*z* 709.21 [M−H]^−^). In addition, several nodes clustered with the discriminatory nodes belonging to cluster 1 were *m*/*z* 623.635, *m*/*z* 626.43, *m*/*z* 664.939, *m*/*z* 725.21, and *m*/*z* 767.22, which could not be purified in sufficient quantity to allow chemical identification. These nodes identified by GNPS provided no hits, which strongly suggests the presence of additional analogs in this chemotype.

Other nodes, such as (singleton) at an average *m*/*z* of 599.1747 (compounds **1**/**2**) and *m*/*z* of 617.1865 (compound **4**), that had no matches in the GNPS database were attributed to dimeric xanthone derivatives. Compounds **1**, **2**, and **4** did not cluster with dimeric xanthone in cluster 1 because the product ions displayed a similarity score (cosine score) of <0.65. Subsequently, purification of the *m*/*z* 599.1759 and metabolites led to a pair of configurational xantone dimers named here Phomoxanthones L (**1**) and M (**2**), and *m*/*z* phomoxanthone D (**4**). In addition, we tabulated the structures of some nodes within clusters 1–10 annotated from molecular networking in Appendix A.

Thereafter, from a careful metabolomics analysis (PCA, HPLC chromatogram, ^1^HNMR spectroscopy, and molecular networking) of the co-cultivation, the targeted dimeric xantone derivatives including three new induced compounds were prioritized for isolation and chemical characterization.

### 2.3. Structure Elucidation of New Compounds

Phomoxanthone L (**1**) was isolated as a light-yellow amorphous powder, and its molecular formula was established as C_30_H_30_O_13_ based on HRESIMS data (*m*/*z* 599.1755 [M+H]^+^, calcd for C_30_H_31_O_13_ 599.1765), indicating 16 indices of unsaturation. The ^1^H and ^13^C nuclear magnetic resonance (NMR) data (Table 1) of **1** in association with distortionless enhancement by polarization transfer (DEPT) and heteronuclear single quantum coherence (HSQC) spectrum showed the presence of a series of characteristic signals for two secondary methyls [*δ*_H_ 1.12, (d, *J* = 6.8 Hz), *δ*_C_ 14.7, q, 11-CH_3_; 1.09, (d, *J* = 7.1 Hz), *δ*_C_ 19.6, q, 11′-CH_3_], two methylenes [*δ*_H_ 2.09 (d, *J* = 13.8 Hz), *δ*_H_ 1.69 (dd, *J* = 13.8, 5.8 Hz), *δ*_C_ 37.1, t, CH_2_-7; *δ*_H_ 2.17 (dd, *J* = 17.7, 9.0 Hz), *δ*_H_ 1.95 (dd, *J* = 17.7, 9.2 Hz), *δ*_C_ 35.6, t, CH_2_-7′], two methines (*δ*_H_ 2.33, m, *δ*_C_ 29.2, d, CH-6; *δ*_H_ 2.61, m, *δ*_C_ 28.7, d, CH-6′), two oxygenated methylenes [*δ*_H_ 3.89 (d, *J* = 9.7 Hz), *δ*_H_ 3.83 (d, *J* = 9.7 Hz), *δ*_C_ 64.6, t, CH_2_-12; *δ*_H_ 3.82 (d, *J* = 11.6 Hz), *δ*_H_ 3.71 (d, *J* = 11.6 Hz), *δ*_C_ 61.0, t, CH_2_-12′], two oxygenated methines [*δ*_H_ 4.11 (d, *J* = 4.1 Hz), *δ*_C_ 76.4, d, CH-5; *δ*_H_ 4.30 (d, *J* = 3.4 Hz), *δ*_C_ 87.4, d, CH-5′], four adjacent olefinic methines [*δ*_H_ 7.47 (d, *J* = 8.5 Hz), *δ*_C_ 141.8, d, CH-3; *δ*_H_ 6.62 (d, *J* = 8.5 Hz), *δ*_C_ 107.4, d, CH-4; *δ*_H_ 7.36 (d, *J* = 8.5 Hz), *δ*_C_ 139.6, d, CH-3′; *δ*_H_ 6.52 (d, *J* = 8.5 Hz), *δ*_C_ 108.6, d, CH-4′], and 15 quaternary carbons (including two conjugated carbonyl at *δ*_C_ 199.76 (C-9) and *δ*_C_ 197.94 (C-9′)), and one ester carbonyl at *δ*_C_ 177.3 (C-8′)]. These spectroscopic data suggest that **1** was a heterodimer, with one monomeric moiety similar to a monomeric unit of **4** and the other identical to that of **5**, which were also isolated from the culture medium. The heteronuclear multiple bond correlation (HMBC) correlations (Figure 5) from H-3/C-2′ and H-3′/C-2 deduced the connection of two monomeric moieties should be connected via 2,2′- linkage and, consequently, permitted the construction of the planar structure of **1**. The relative configuration of **1** was established by analyzing the NOESY spectrum on the basis of the monomeric moieties. The NOE cross-peaks (Figure 6) between H-5/H-6, H-5/H_2_-12, H-6/H_2_-12, and H-6/H_a_-7 revealed their co-facial relationship and were arbitrarily assigned as α-orientations, whereas those between H_3_-11 and H_b_-7 indicated that these protons have β orientations in the **4** moiety. The NOE interactions of H-5′/H_3_-11′/H_b_-7′ and H-5′/H_2_-12′ suggested that they were positioned on the same face, and the correlations between H-6′ and H_a_-7′ revealed that they were positioned on the other face, which was further confirmed by the similar coupling constants of these protons in **5**. Since compound **1** was co-isolated with **4**–**6** in our study, it was expected also to contain a monomer with 5*R*, 6*R*, 8*R*, 8a*S*, and 10a*R* absolute configuration on the basis of biogenetic considerations. Accordingly, four diastereomeric starting structures with (5*R*, 6*R*, 8*R*, 8a*S*, 10a*R*, 5′*R*, 6′*R*, 10a′*S*; 5*S*, 6*S*, 8*S*, 8a*R*, 10a*S*, 5′*S*, 6′*S*, 10a′*R*; 5*R*, 6*R*, 8*R*, 8a*S*, 10a*R*, 5′*S*, 6′*S*, 10a′*R*; 5*S*, 6*S*, 8*S*, 8a*R*, 10a*S*, 5′*R*, 6′*R*, 10a′*S*) configuration of **1** and its diastereomer **2** were subjected to calculation of the electronic circular dichroism (ECD) spectrum at the B3LYP/6-31 + g (d, p) level to compare with the experimental circular dichroism (CD) spectrum (Figure 7). Therefore, the absolute configuration of **1** was assigned to 5*R*, 6*R*, 8*R*, 8a*S*, 10a*R*, 5′*R*, 6′*R*, and10a′*S*, and the trivial name phomoxanthone L was ascribed.

Phomoxanthone M (**2**) possesses the same molecular formula of C_30_H_30_O_13_ as **1**. Its NMR data (Table 1) and ^1^H–^1^H COSY, HMBC (Figure 5), and NOESY (Figure 6) correlations were very similar, and the planar structure was elucidated to be the same as that of **1**. The differences between the ^1^H and ^13^C NMR chemical shifts of CH-5′ [*δ*_H_ 4.43 (d, *J* = 4.0 Hz), *δ*_C_ 88.6, d, CH-5′], CH-6′ (*δ*_H_ 2.86, m, *δ*_C_ 31.0, d), CH_2_-7′(*δ*_H_ 2.88, m, *δ*_H_ 2.28, m, *δ*_C_ 37.3, t) and 11′-CH_3_ [1.23, (d, *J* = 5.9 Hz), *δ*_C_ 20.8, q] were larger than those between the other portions in **2** and those of **1**, thus suggesting that **2** might be a diastereomer of **1** at the β-methyl-γ-lactone ring. The absolute stereochemistry of phomoxanthone M (**2**) was ultimately established by means of comparing the experimental and calculated ECD spectra using TDDFT (Figure 7), and were determined as 5*R*, 6*R*, 8*R*, 8a*S*, 10a*R*, 5′*S*, 6′*S* and 10a′*R*. Thus, the structure of **2** was determined, and it was named phomoxanthone M.

The molecular formula of phomoxanthone N (**3**) was determined to be C_30_H_32_O_13_ by the positive HRESIMS ion at *m*/*z* 601.1916 [M+H]^+^ (calcd for C_30_H_33_O_13_ 601.1921), implying 15 indices of unsaturation. A detailed analysis of the1D and 2D NMR spectral data of **3** revealed its feature of the asymmetric tetrahydroxanthone-chromanone heterodimer, and the planar structures of the two segments were the same as those of chromanone monomer of **2** and phaseolorin D, independently. In the same way, on the basis of the H-C long-range correlations (Figure 5) of H-3/C-2′ and H-3′/C-2 and the NMR values for the two linked benzene rings, it was illustrated that the connection of two monomeric moieties has the same 2,2′-linkage as **1** and **2**. Key NOESY correlations (Figure 5) of H-3/C-2′ and H-3′/C-2, as well as the NMR values for the two linked benzene rings illustrated the connection of two monomeric moieties has the same 2,2′-linkage as **1** and **2**. Key NOESY correlations (Figure 6) of H-5/H_3_-11, H_3_-11/H-8, and H-8/8a-OH suggesting that these protons are co-facial and have β-orientations, correlations between H-6/H_2_-12, and the absence of correlations between H-8/H-6 and 8a-OH/H_2_-12 indicate the α-orientations of protons H-6 and H_2_-12. In addition, the almost identical NMR data and NOESY correlations as described for **2** confirmed the absolute configuration of the chromanone unit in **3**. ECD calculations (Figure 7) were further employed to determine the absolute configuration. As shown, the 5*S*, 6*R*, 8*R*, 8a*S*, 10a*R*, 5′*S*, 6′*S*, and 10a′*R* configuration was defined based on the calculated ECD spectrum in good accordance with the experimental curve.

### 2.4. Biological Evaluation of Isolated Compounds

All the isolated metabolites were evaluated for their cytotoxic and immunosuppressive activities according to the previously described methods [46,47,48]. Compounds **1**–**9** showed broad cytotoxic activity (Table 2). Amongst these, compounds **7**–**9** exhibited significant cytotoxicity towards human liver cells HepG-2 (IC_50_ values of 4.83 ± 0.22, 13.99 ± 1.13, and 12.06 ± 0.55 μM) and cervical cancer cells Hela (IC_50_ values of 18.96 ± 0.88, 23.42 ± 2.55, and 20.36 ± 1.99 μM). The results indicate that the tetrahydroxanthone moieties and hydroxyl groups attached at C-12 and C-12′ were the key functional architectures contributing to the cancer cells’ proliferation inhibitory effect. Compounds **1**–**6** exhibited weak immunosuppressive activity against the proliferation of ConA-induced (T-cell) and LPS-induced (B-Cell) murine splenic lymphocytes (Table 3).

## 3. Materials and Methods

### 3.1. General Procedures

The optical rotations were acquired using the ATR-W2 HHW5 digital Abbe refractometer (Shanghai Physico-optical Instrument Factory, Shanghai, China). The UV spectra were obtained using a Shimadzu UV-2600 PC spectrophotometer (Shimadzu Corporation, Tokyo, Japan), while the ECD spectra were measured on a JASCO J-715 spectra polarimeter (Japan Spectroscopic, Tokyo, Japan). All the LC/MS data were collected by an LCMS-IT-TOF instrument (Shimadzu Corporation, Tokyo, Japan) with an ESI source. The ^1^H, ^13^C, and 2D NMR spectra were acquired on a Bruker AV 400 NMR spectrometer using TMS as an internal standard. TLC and column chromatography (CC) were executed on silica gel (200–400 mesh, Qingdao Marine Chemical Inc., Qingdao, China) or a Sephadex-LH-20 (18−110 µm, Merck, Darmstadt, Germany), respectively. The HPLC analysis was measured on Wasters e2695 (Waters Corporation, Milford, MA, USA) using a C18 column (Waters, 5 μm, 10 × 150 mm). Semi-preparative HPLC was obtained on an Agilent Technologies 1200 LC with a C18 column (Agilent Technologies 10 mm × 250 mm). High-speed centrifugation was performed on a TGL-16B Anting centrifugal machine (Anting Scientific instrument Factory, Shanghai, China). The constant temperature water bath was in HH-2 thermostat water baths (Hervey Biotechnology Corporation, Jinan, China). The purity of the isolated compounds was determined via high-performance liquid chromatography (HPLC), which was performed on an Agilent 1200 instrument and a reverse-phase column (4.6 × 150 mm, 5 μm). The UV wavelength for detection was 210 nm. All the crude extracts and compounds were eluted with a flow rate of 0.8 mL·min-1 over a 50 min gradient (solvents: A, H_2_O; B, MeOH), as follows: 0–5 min, 25% B; 5–15 min, 25–30% B; 15–30 min, 30–55% B; 30–40 min, 55–75% B; 40–50 min, 70–90% B; 50–60 min, 90–100% (Appendix A).

### 3.2. Fungal Material

The endophytic fungi Phomopsis asparagi and *Phomopsis* sp. were isolated from the fresh root of the mangrove plant Rhizophora mangle collected in Dong Zhai Gang-Mangrove Garden on Hainan Island, China, in October 2015. The fungi were identified as *Phomopsis asparagi* (strain no.DHS-48) and *Phomopsis* sp. (strain no.DHS-11) by ITS gene sequence (GenBank Accession No.MT126606 and No. OR801625). Two voucher strains were deposited at one of the authors’ laboratories (J.X.).

### 3.3. Preparation of Phomopsis asparagi, Phomopsis sp., Co-Cultivation, and Morphological Observation

The two fungi, *Phomopsis asparagi* and *Phomopsis* sp., were cultured independently on PDA for 7 days at 28 °C. For co-cultivation, two 1 cm^2^ pieces of agar from each fungus were placed 5 cm distance from each other on a new agar plate (9 cm in diameter). The co-cultures were incubated at 28 °C for 14 days. In parallel, mono-cultures of each strain used for co-cultivation were prepared and cultivated in the same conditions for comparison. After 14 days, the agar blocks with mycelium of *Phomopsis asparagi*, *Phomopsis* sp., and their co-culture were cut and mixed with 2.5% glutaraldehyde solution at 4 °C for 4 h and meticulously rinsed in phosphate buffered saline solution (PBS) three times for 15 min each. The segments were dehydrated three times in graded ethanol in a series (50%, 70%, 80%, 90%, and 100%) over a period of time for 30 min each time, then lyophilized, fixed on radio with conductive gel, sprayed with gold for 30 s with an IB-3 ion coating apparatus, and, finally, observed by the Hitachi S-3000N scanning electron microscope.

### 3.4. Sample Preparation of Phomopsis asparagi, Phomopsis sp., Co-Culture, and Large-Scale Fermentation and Extracts

The two fungi were independently cultivated on PDA at 28 °C for 14 days. After that, the two fungi colonies were simultaneously inoculated into an autoclaved rice solid-substrate medium in Erlenmeyer flasks (130 × 1 L); each contained 100 g of rice and 100 mL of 0.3% saline water and were fermented at 28 °C for 30 days. At the same time, *Phomopsis asparagi* and *Phomopsis* sp. were separately cultured under the same culture conditions in 20 Erlenmeyer flasks.

Following the fermentation process, a random selection of 5 bottles was made from the 130 co-cultured fermentation mixes, and similarly, 5 bottles were randomly chosen from the 20 bottles of *Phomopsis asparagi* and *Phomopsis* sp. monocultured fermentation mixes. The co-cultured and monocultured fermentation mixes that were selected were extracted three times with EtOAc, and the filtrate was then distilled under reduced pressure to obtain the crude extracts. The operations outlined above are to ensure that all the co-cultures and monocultures have five independent biological replicates, ensuring the reliability and of the results. The dried extracts were re-dissolved in an appropriate amount of UPLC/MS-grade methanol (MeOH) and pipetted into a pre-weighed 1.5 mL-amber glass vial through a 13 mm syringe filter with a 0.22 mm PTFE membrane to prepare a 1 mg·mL^−1^ solution for future use. The crude extracts were analyzed using HPLC, UPLC-MS/MS, and ^1^H NMR. The remaining 125 bottles of the co-cultured fermentation mixes were similarly extracted three times with EtOAc to obtain 30 g of crude extract for subsequent isolation.

### 3.5. UPLC-ESI-MS/MS Analysis

The combination of the Japanese Shimazu liquid phase system and mass spectrometry system was used to accomplish the UPLC-ESI-MS/MS analysis. To keep the instrument in the best possible condition before analysis, the chromatographic column and instrument system have to be cleaned with the appropriate solvents. The samples were injected and chromatographically separated on a Waters ACQUITY UPLC BEH CI8 column (10 mm × 210 mm, 1.7 μm) at a temperature of 25 °C with an injection volume of 5 μL and the PDA detector set at 190–400 nm. The mobile phase adopted a binary elution system, with the A phase containing water and the B phase containing methanol. The elution procedure is 0–5 min, 25% B; 5–15 min, 25–30% B; 15–30 min, 30–55% B; 30–40 min, 55–75% B; 40–50 min, 70–90% B; 50–60 min, 90–100%, with a flow rate of 0.3 mL·m^−1^.

The ESI ion source parameters were set as follows: sensitivity mode under positive and negative ions; ionization voltage, 2.00 kV; cone hole voltage, 40 V; ion source voltage compensation, 80 V; ion source temperature, 100 °C; temperature, 250 °C; cone hole gas flow rate, 50 L/h; flow rate, 600 Lh; the and cone-shaped gas were high-purity nitrogen. The scanning ion range was *m*/*z* 100–2000 Da, and the energy setting was 2 V for low energy and 40–80 V for high energy; the scanning time was 0.10 s. An MS survey scan was performed in the range of 100–1500 Da, while MS/MS scanned over a mass range of 50–1500 Da at the same time. As controls, the solvent (MeOH) and non-inoculated medium were injected under the same conditions. The data were collected and analyzed using the LCMSsoultion Ver. 3 software (Shimazu, Japanese).

### 3.6. Data Processing, Molecular Networking, Dereplication, and Multivariate Data Analysis

After the collection was completed, MSConvert was used to convert the data into mzXML format and construct a molecular network in GNPS (Global Natural Products Social Molecular Network “http://gnps.ucsd.edu (accessed on 4 February 2024)” [59]. The MS/MS molecular network was constructed using the classic online workflow (METABOLOMICS-SNETS-V2) in GNPS. The parameters were set to at least four matching peaks, a minimum cosine similarity score of 0.65, a parent mass tolerance of 1.0 Da, and a fragment ion tolerance of 0.3 Da. The spectra in the network were then searched against GNPS spectral libraries to annotate and identify metabolites through the database search of the MS/MS spectra. The MN work on GNPS can be found at http://gnps.ucsd.edu/ProteoSAFe/status.jsp?task=4ae8b81709dc4f3899d56a9e2f185f10, accessed on 4 February 2024. The resulting MN was visualized using Cytoscape 2.8.3 for composition and displayed with ‘directed’ style. The thickness of the edges between the nodes is adjusted by the cosine value, and the color of the nodes is determined based on the source of the spectral file. To simplify the analysis of the network, background nodes originating from the cultivation medium rice and solvents (MeOH) were removed from the MN [60]. Node colors were mapped based on the source of the spectra files, as *Phomopsis asparagi* are green, *Phomopsis* sp. are yellow, and the nodes of the co-culture are red. To improve the dereplication, a manual annotation was conducted in LCM solution Ver. 3 software by searching the microbial natural products database, such as the Dictionary of Natural Products (DNP) “http://dnp.chemnetbase.com (accessed on 5 February 2024)”, the Natural Products Atlas “www.npatlasorg (accessed on 5 February 2024)”, and the NPASS “http://bidd.group/NPASS (accessed on 5 February 2024)”. Based on the predicted accurate mass value, the error range is within the target MW ± 1 Da, UV features, MS/MS spectrum, and taxonomic unit information (mainly within the *Phomopsis* sp. And, if necessary, extended to the fungal kingdom) [27,56,61,62,63]. All the data have been uploaded to the MassIVE Database of the GNPS Web site “https://gnps.ucsd.edu/ProteoSAFe/static/gnps-splash2.jsp (accessed on 15 February 2024)” and are publicly available through access number MSV000094093. Dereplication was also achieved by isolation and characterization by NMR and HR-ESIMS.

The HR-MS/MS data were analyzed using Mzmine 2.5.3. Markers between 150 and 1500 Da were collected with an intensity threshold of 10k counts and the retention time and mass windows of 0.2 min and 0.1 Da, respectively. The noise level was set to 3.0 × 10^3^, and the raw data were deisotoped. A statistical analysis of the data was conducted using SIMCA (version 14.0, Umetrics, Umea, Sweden) [28]. With the parameters applied for the collection of markers, approximately 300 separate markers for each survey were identified in total. The markers from the extracts were compared through PCA.

### 3.7. Isolation of Compounds

After the completion of fermentation, the co-cultured fermentation mixes were extracted three times with EtOAc, and the filtrate was then distilled under reduced pressure to obtain 30 g of crude extract. Using stepped gradient elution with CH_2_Cl_2_-MeOH (0–100%) on silica gel column chromatography (CC), the crude extracts were separated into nine fractions (Fr. 1–Fr. 9). The fraction Fr. 3 was subjected to open silica gel CC using gradient elution with CH_2_Cl_2_-MeOH-100:0–1:1, *v*/*v* to obtain 6 fractions (Fr. 3.1–Fr. 3.6). Fr. 3.3 was chromatographed on a Sephadex LH-20 CC by eluting with MeOH to yield compounds **7** (20 mg) and **9** (500 mg). Fr. 4 was subjected to open silica gel CC using gradient elution with CH_2_Cl_2_-MeOH-100:0–1:2, *v*/*v* to obtain 7 fractions (Fr. 4.1–Fr. 4.7). The subfraction Fr. 4.3 was applied to ODS CC with a gradient elution of MeOH/H_2_O mixtures (*v*/*v*, 1:4, 3:7, 2:3, 1:1, 3:2, 7:3, 4:1, 0:1) and obtained five subfractions (Fr. 4.3.1–Fr. 4.3.5). Then, Fr. 4.3.4 was purified by semi-preparative reversed-phase HPLC using MeOH-H_2_O (60:40, *v*/*v*, 2 mL·m^−1^, UV λ_max_ 210 nm) to afford compounds **6** (10 mg), **1** (5 mg), and **5** (7 mg), and Fr. 4.3.5 was purified by semi-preparative reversed-phase HPLC using MeOH-H_2_O (60:40, *v*/*v*, 2 mL·m^−1^, UV λ_max_ 210 nm) to afford compound **2** (10 mg) and compound **8** (10 mg). Fr. 7 was subjected to open silica gel CC using gradient elution with CH_2_Cl_2_-MeOH-100:2–1:2, *v*/*v* to obtain 5 fractions (Fr. 7.1–Fr. 7.5). Fr. 7.2 was chromatographed on a Sephadex LH-20 CC by eluting with MeOH to yield three fractions (Fr. 7.2.1–Fr. 7.2.3). Fr. 7.2.3 was purified by semi-preparative reversed-phase HPLC using MeOH-H_2_O (60:40, *v*/*v*, 2 mL·m^−1^, UV λ_max_ 210 nm) to afford compound **3** (6 mg). Fr. 8 was applied to ODS CC with gradient elution of MeOH/H_2_O mixtures (*v*/*v*, 1:4, 3:7, 2:3, 1:1, 3:2, 7:3, 4:1, 0:1) and obtained five subfractions (Fr. 8.1–Fr. 8.5). Fr. 8.2 was chromatographed on a Sephadex LH-20 CC by eluting with MeOH to yield three fractions (Fr. 8.2.1–Fr. 8.2.3). Fr. 8.3 was purified by HPLC (MeOH/H_2_O, 70:30 and 60:40, *v*/*v*; 2 mL·m^−1^, UV λ_max_ 210 nm) to yield compound **4** (8 mg).
Phomoxanthone L (**1**): yellow amorphous powder (MeOH); [α]^20^_D_ −20 (c 0.0001, MeOH); UV (MeOH) λ_max_ 205, 251, 361 nm (the absorptions due to aromatic rings); ^1^H and ^13^C NMR data, see Table 1; HRESIMS *m*/*z* 597.1613 [M − H]^−^ (calcd for C_30_H_29_O_13_ 597.1614).Phomoxanthone M (**2**): yellow amorphous powder (MeOH); [α]^20^_D_ −10 (c 0.0001, MeOH); UV (MeOH) λ_max_ 207, 254, 364 nm (the absorptions due to aromatic rings); ^1^H and ^13^C NMR data, see Table 1; HRESIMS *m*/*z* 599.1759 [M+H]^+^ (calcd for C_30_H_31_O_13_ 599.1759).Phomoxanthone N (**3**): yellow amorphous powder (MeOH); [α]^20^_D_ +10 (c 0.0001, MeOH); UV (MeOH) λ_max_ 212, 255, 366 nm (the absorptions due to aromatic rings); ^1^H and ^13^C NMR data, see Table 1; HRESIMS *m*/*z* 601.1916 [M+H]^+^ (calcd for C_30_H_33_O_13_ 601.1916).

### 3.8. Theory and Calculation Details

Detailed Monte Carlo conformational analyses were performed utilizing Spartan’s 14 software and the Merck molecular force field (MMFF). Conformers exceeding a Boltzmann population of 0.4% were selected for electronic circular dichroism (ECD) calculations as presented in Appendix A. Subsequently, these conformers underwent initial optimization at the B3LYP/6-31G(d) level in the gas phase, complemented by the polarizable conductor calculation model based on the polarizable continuum model (PCM). The stable conformations identified at the B3LYP/6-31G(d) level were then used in magnetic shielding constants. The theoretical calculation of ECD was conducted in MeOH using the time-dependent density functional theory (TD-DFT) at the B3LYP/6-31 + g (d, p) level for all the conformers of compounds **1**, **2**, and **3**. The ECD spectra were generated with the aid of the SpecDis 1.6 program (University of Würzburg, Würzburg, Germany) and GraphPad Prism 5 (University of California, San Diego, CA, USA) through the conversion of dipole-length rotational strengths into band shapes modeled by Gaussian functions with a standard deviation of 0.3 eV.

### 3.9. Cytotoxicity Assay

The liver cancer cell line, HepG2, and the cervical cancer cell line, Hela, were obtained from the Type Culture Collection of the Chinese Academy of Sciences in Shanghai, China. The cells were cultivated using RPMI-1640 culture medium. To assess cytotoxicity against HepG2 and HeLa cells, the 3-(4,5-dimethylthiazol-2-yl)-2,5-diphenyltetrazolium bromide (MTT) method, sourced from Sigma-Aldrich in St. Louis, Missouri, USA, was employed as described previously [47]. Furthermore, adriamycin (from Shanghai Macklin Biochemical Co., Ltd., with a purity of 99.8%) (Shanghai, China) and 5-fluorouracil (5-FU) (from Beijing Solarbio Science and Technology Co., Ltd., with a purity of 99.8%) (Beijing, China), both served as positive controls.

### 3.10. Splenocyte Proliferation Assay

Spleen cells were collected from BALB/c mice under aseptic conditions, plated in a 96-well plate at a concentration of 1 × 10^7^ cells/mL per well, and activated by Con A (5 μg/mL) or LPS (10 μg/mL) in the presence of various concentrations of the compounds or cyclosporine A (CsA) at 37 °C and 5% CO_2_ for 48 h. Then, 20 μL CCK-8 was added to each well 4 h before the end of the incubation. The absorbance at OD_450_ was measured on an ELISA reader, and the IC_50_ value was calculated from the correlation curve between the compound concentration and the OD_450_.

### 3.11. Statistical Analysis

All the cell data are presented as the mean standard deviation of the means (S.D.), and a one-way analysis of variance (ANOVA) was used to evaluate the statistical significance of the differences between the groups by GraphPad Prism.

## 4. Conclusions

Mangrove endophytic fungi are considered one of the most promising sources of novel biologically active compounds. In the current study, co-cultivation of mangrove *Rhizophora mangle* endophytic fungi *Phomopsis asparagi* DHS-48 and *Phomopsis* genus fungus DHS-11 led to global changes of their metabolic profiles. Further analysis using the metabolomic approach integrated HPLC chromatogram, ^1^HNMR spectroscopy, UPLC-MS-PCA, and molecular networking, indicating the presence of a series of induced dimeric xantones, resulting in the targeted isolation and structure elucidation of three new phomoxanthones L-N (**1**–**3**) and six known analogs (**4**–**9**). Meanwhile, compounds **7**–**9** significantly suppressed the proliferation of human liver cells HepG-2. Our study highlights that combining co-cultivation with a metabolomics-guided strategy as a discovery tool will be implemented as a systematic strategy for the quick discovery of target bioactive compounds.

## Data Availability

The original data presented in the study are included in the article/Appendix A; further inquiries can be directed to the corresponding author.

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
