# Peer review of "Metabolomics-Guided Discovery of New Dimeric Xanthones from Co-Cultures of Mangrove Endophytic Fungi Phomopsis asparagi DHS-48 and Phomopsis sp. DHS-11"

_marinedrugs, 2024, doi:10.3390/md22030102_

Round 1

Reviewer 1 Report

Comments and Suggestions for Authors

This paper reports on the results of co-culturing the two mangrove endophytic fungi Phomopsis asparagi DHS-48 and Phomopsis sp. DHS-11, a well-known strategy for developing the production of novel metabolites. Morphological and scanning electron microscopic observations as well as HPLC profiles suggested the production of modified metabolites in the co-culture extract. The LC-MS/MS experiments of each extract were combined with molecular network building and the use of spectral libraries for node annotation to explore the chemical diversity of co-culture and monoculture extracts. No nodes were annotated from the co-culture clusters, but six known dimeric xanthones (4-9) were annotated from the co-culture nodes and their respective monocultures, allowing the identification of three new analogs (1-3). After isolation and purification, the three new dimeric xanthones were identified as being phomoxanthones (1-3) by detailed analysis of 1D and 2D NMR data, and computational calculations of ECD spectra determined their stereostructures.

All isolated dimeric xanthones (1-9) were evaluated for their cytotoxic and immunosuppressive activities.  The known dimeric xanthones (7-9) showed significant cytotoxic activity with IC50 values ranging from 4.83 to 12.06 µM against the liver cancer cell line HepG2, whereas only a weak immunosuppressive activity was observed for xanthones (1-6) against the proliferation of Con-A-induced (T cell) and LPS-induced (B cell) murine splenic lymphocytes. 

This is an interesting study.  However, it is not clear what tool was used to identify the new dimeric xanthones? What do you mean by “Further analysis” (line 233)? 

The authors claim that metabolomics guided the discovery of dimeric xanthones. Metabolomics is defined as a combination of multiparametric chemical analysis techniques and multivariate statistical analyses.

If metabolomics guided the identification of the new dimeric xanthones, what was the procedure (including statistical analysis)? Which variables were used? How many of them? How were they selected? Are they from independent samples?

Multivariate analysis (PCA) was indeed carried out and statistical analyses using SIMCA were performed on the UHPLC-HRMS data, but not to guide the discovery of the new dimeric xanthones, which were identified after isolation and detailed study of their NMR data.

Please, clarify these questions and be more precise as it is too confusing. It would be interesting to see the workflow of the study for a better understanding. I suggest adding a figure to explain the scientific process.  

Also, the caption of Figure S2 should be clarified. It is surprising to see that there is no variability within the three groups of biological models. Please, explain this figure as well. 

Why are the three new dimeric xanthones not in the same cluster? Have you changed the filter parameters for the building of the MN? Have you tried the cosine value of 0.6?  

Lines 186-188: Did you use in silico databases (Moldiscovery, MolNetEnhancer…) to get additional information about the red nodes?   

In the materials and methods section, you mention DNP, but also the Natural Products Atlas, NPASS databases, which are not mentioned in the main text. What information did you get from these tools? Which nodes in the MN were annotated with these tools? Please, explain. 

Figure 4 is not clearly visible. Please, enlarge it. 

Have the MS/MS spectra of the new dimeric xanthones been deposited in the GNPS spectral library? Please, add the identifier CCMSLIB code. 

Please, make these minor following corrections:

Throughout the manuscript, please:   -

            - use the correct italic spelling for “Phomopsis asparagi” and “Phomopsis sp.” (with “sp” not in italics)

- write “m/z” is in italics

- write “mL.m-1” and “mg.mL-1” (use the superscript and not the spelling  “mL/min” and “mg/mL”)

- check that the underscore is always used for “H2O”

- check that the superscript is always used for “1H” and “13C”

- check and add some spaces between some numbers and their unit (e.g. page 13, line 451 “30g”) 

In addition,

In the abstract, lines 11-12: “…strategy, which mimics…community, is a useful…”

In the abstract, line 13: “…to increase the yield of…”

In the abstract, line 16: “…DSH-11, both…”

In the abstract, line 23: “…Their planar structures…”

Page 1, lines 36-41: This sentence is too long. Please, shorten it.

Page 2, line 36: “…1,300 new compounds” and page 13, line 442: “1,500 Da”

Page 2, line 63: “There are several insights…”

Page 2, line 66: “…phenols…”

Page 5, line 159; “resonations” should be “resonances”

Page 6, line 198: “…two nodes are numbered…”

Page 7, line 234: “…as well as comparison…”

Page 7, lines 233-234: Please, rephrase.

Page 9, Table 1: Please, check the character for compound 3 dC, 9a

Page 13, line 433: Please, correct “LCMSoultion Ver3”

Page 15, lines 517: Please, rewrite the sentence  

References: The dois are missing 

Comments on the Quality of English Language

English language is fine.

Author Response

Response to Reviewer :

Comments and Suggestions for Authors

Question 1: This is an interesting study.  However, it is not clear what tool was used to identify the new dimeric xanthones? What do you mean by “Further analysis” (line 233)?

Answer: Thank you very much for your positive comment! By the analysis of the MN of co-culture and respective monocultures crude extracts, we first identified a series of dimeric xantone compounds (known xantone dimmers and the new compound 3) in cluster 1, which gave us great interest in further studying and identifying a series of similar compounds. “Further analysis” refers to the subsequent in-depth analysis of the molecular network of other clusters except cluster 1, a singleton node represent compounds 1 and 2 at average m/z 599.1747, another singleton node represent compound 4 at m/z 617.1865,  had no matches in the GNPS database. These nodes were not in the same cluster as the nodes in cluster 1, because their cosine values were less than 0.65.

Question 2: The authors claim that metabolomics guided the discovery of dimeric xanthones. Metabolomics is defined as a combination of multiparametric chemical analysis techniques and multivariate statistical analyses.

If metabolomics guided the identification of the new dimeric xanthones, what was the procedure (including statistical analysis)? Which variables were used? How many of them? How were they selected? Are they from independent samples?

Multivariate analysis (PCA) was indeed carried out and statistical analyses using SIMCA were performed on the UHPLC-HRMS data, but not to guide the discovery of the new dimeric xanthones, which were identified after isolation and detailed study of their NMR data.

Please, clarify these questions and be more precise as it is too confusing. It would be interesting to see the workflow of the study for a better understanding. I suggest adding a figure to explain the scientific process.

Also, the caption of Figure S2 should be clarified. It is surprising to see that there is no variability within the three groups of biological models. Please, explain this figure as well.

Answer: Thank you very much for your constructive suggestions! Metabolomics is defined as a combination of multiparametric chemical analysis techniques and multivariate statistical analyses. We first use (1) HPLC chromatograms followed by (2) LC−MS-based principal component analysis (PCA) to distinguish whether metabolomic changes engendered in the co-culture and monoculture samples based on their UPLC-MS profiles. Then we use (4) 1H NMR-based approaches to inspect the structural features of specific chemotypes based on the chemical shift in 1H NMR spectrum to discriminate alternative candidate structures in the dereplication process. Finally we adopted (4) GNPS platform to create molecular networking based on UPLC-MS/MS fragmentation patterns for dereplication and offer information about the chemical diversity in selected extracts before isolation. So as mentioned in the abstract and conclusion of the manuscript, a new approach which integrates HPLC chromatogram, 1HNMR spectroscopy, UPLC-MS-PCA, and molecular networking enabled the successfully targeted isolation of the induced dimeric xantone metabolites. For statistical analysis, UPLC-ESI-MS were conducted five times to acquire more accurate PCA results under the identical conditions for coculture and monoculture crude extracts. All collected UPLC-ESI-MS data were converted into mzXML format using MS Convert software. Data pre-processing and statistical analysis were performed with MZmine 2.5.3. The statistical analysis was carried out using SIMCA software. In the process of data preprocessing, the peak detection threshold for MS signal intensity was set to 10k. The chromatogram building was realized using a minimum time span of 0.01 min, the noise level was set to 3 × 103, and m/z tolerance of 0.005 (or 10 ppm). Chromatograms were deconvoluted with the following settings: search minimum in absolute retention time (RT) range 0.2 min, minimum relative height 10%, minimum absolute height 2.5 × 103 and baseline level 1.2. The chromatogram isotopic peaks grouper algorithm was set as m/z tolerance of 0.005 (or 10 ppm) and absolute RT tolerance of 0.2 min. Chromatograms were peak aligned with m/z tolerance at 0.008 (or 15 ppm) and absolute RT tolerance 1 min. The peak list was eventually gap-filled with m/z tolerance at 0.008 (or 15 ppm), and absolute RT tolerance of 0.20 min. Each sample was preprocessed and obtain 600 peaks, in order to classify these peaks, principal component analysis (PCA) was carried out by SIMCA. The Data was then analyzed by PCA analysis using SIMCA software. First, Set the m/z as the Primary ID and RT as the Secondary ID. Then normalization of the variables with Pareto scaling. After that, PCA were displayed by the scores plot to observe the overall cluster. By observing the score plots of co-culture and its monocultures, we found that the five replicate groups of DHS-48 monoculture were closely integrated, the five replicate groups of DHS-11 monoculture were closely integrated, and the five replicate groups after their co-culture were integrated, meanwhile the three types of biological samples were separated from each other obviously, indicating that their chemical compositions were different. In addition, there is no variability within the three groups of biological models because we performed the identical treatment to each biological sample and collected data five times under the same circumstances, to demonstrate the stability of the results. In order to clarify the process, we added the sentence “1H NMR-based approaches has shown to be very effective in discriminating between alternative candidate structures in the dereplication process. It can provide comprehensive characteristic fingerprints since the co-cultivation can greatly affect the fingerprints of the extracts, based on structural features of specific chemotypes that can be easily determined from inspection of the chemical shift in 1H NMR spectrum [29]” in line 81-86. We added score plots besides the loading plot in Figure S2A and S2B of supp. Info., and the parameters were labled out in the figure annotation. Finally, in order to clarify these questions and be more precise, we replaced the graphical abstract to include the workflow of the study for a better understanding according to your suggestion!

Question 3: Why are the three new dimeric xanthones not in the same cluster? Have you changed the filter parameters for the building of the MN? Have you tried the cosine value of 0.6?

Answer: Thank you very much for your suggestions! For the molecular network analysis, the parameters are set to at least four matching peaks, a minimum cosine similarity score of 0.65, a parent mass tolerance of 1.0 Da, and a fragment ion tolerance of 0.3 Da. In the network, compounds 1, 2 (m/z 599.1747) and compound 4 (m/z 617.1865) were not in the same cluster as the nodes of compounds 3, 5, 6-9 in cluster 1, because the existence of an additional oxygen bridge formed by CH2-12 in the tetrahydrofuran ring and difference between their structure and the MS/MS fragments with cosine values less than 0.65. We attempted to set the cos to 0.6, but unfortunately, they still did not cluster together in the same cluster. We also consulted literature, which reported, pseudoanguillosporin B (7), that bears a hydroxy substitution on the alkyl side chain, and did not cluster with the isochromans in cluster N because the product ions displayed a similarity score (cosine score) <0.7, as estimated from the MN algorithm in reference “Oppong-Danquah, E.; Blümel, M.; Scarpato, S.; Mangoni, A.; Tasdemir, D. Induction of Isochromanones by Co-Cultivation of the Marine Fungus Cosmospora sp. and the Phytopathogen Magnaporthe Oryzae. Int. J. Mol. Sci. 2022, 23, 782, doi:10.3390/ijms23020782.”. 

Question 4: Lines 186-188: Did you use in silico databases (Moldiscovery, MolNetEnhancer…) to get additional information about the red nodes?

Answer: Thank you very much for your comments and reminding. We conducted a manual annotation using LCMS soultion Ver3 software by searching the microbial natural products database such as the Dictionary of Natural Products (DNP) (http://dnp.chemnetbase.com), the Natural Products Atlas (www.npatlasorg) and the NPASS (http://bidd.group/NPASS) based on the predicted accurate mass value, the error range is within the target MW ± 1 Da, UV features, MS/MS spectrum, and taxonomic unit information. We agreed that databases such sa Moldiscovery, MolNetEnhancer would be useful to get additional information about the red nodes, at this point, we do not have the essential tool-set to get access to these database. We hope, in the future, we could explore more tools to enhance annotation of molecular network nodes to gain more information.

Question 5: In the materials and methods section, you mention DNP, but also the Natural Products Atlas, NPASS databases, which are not mentioned in the main text. What information did you get from these tools? Which nodes in the MN were annotated with these tools? Please, explain.

Answer: Thank you very much for your comments! For the in-depth analysis of the MN biggest cluster 1, we used a series of databases to conduct manual searches and node annotations, such as nodes m/z 667.2021, nodes m/z 583.1811, nodes m/z 709.2129. First, we got the molecular formula from the exact mass data, and then we use this molecular formula to get a search in the databases Natural Products Atlas, DNP and NPASS. We speculated the node structure by comparing the MS/MS fragmentation pattern, UV absorption and biological resource, to speculate the node’s structure. And then, we further confirmed the structure by measure the NMR data of the compounds obtained by the subsequent fermentation separation. Sometimes, a lot of nodes can not be annotated from any database. For instance, node m/z 601.1916 (compound 3), which was proven to be a new compound by the isolated NMR, was not found match result in the database referred to above using the projected molecular formula.

Question 6: Figure 4 is not clearly visible. Please, enlarge it.

Answer: Thank you very much for your comments! We enhanced the resolution response of Figure 4 in the manuscript.

Question 7: Have the MS/MS spectra of the new dimeric xanthones been deposited in the GNPS spectral library? Please, add the identifier CCMSLIB code.

Answer: Thank you very much for your comments! We upload MS/MS spectra to the MassIVE database according to your suggestion! All data have been uploaded to the MassIVE Database atthe GNPS Web site (https://gnps.ucsd.edu/ProteoSAFe/static/gnps-splash2.jsp) and is publicly available through access number MSV000094093.

Question 8: Please, make these minor following corrections:

Throughout the manuscript, please:   -

            - use the correct italic spelling for “Phomopsis asparagi” and “Phomopsis sp.” (with “sp” not in italics) 366-380

- write “m/z” is in italics

- write “mL.m-1” and “mg.mL-1” (use the superscript and not the spelling  “mL/min” and “mg/mL”)

- check that the underscore is always used for “H2O”

- check that the superscript is always used for “1H” and “13C”

- check and add some spaces between some numbers and their unit (e.g. page 13, line 451 “30g”)

Answer: Thank you very much for your suggestions! We are terribly sorry for our mistakes and thank you very much for you carefully inspection. c, we corrected the above spelling mistakes in full text.

Question 9: In addition, In the abstract, lines 11-12: “…strategy, which mimics…community, is a useful…”

In the abstract, line 13: “…to increase the yield of…”

In the abstract, line 16: “…DSH-11, both…”

In the abstract, line 23: “…Their planar structures…”

Answer: Thank you! The comma has been added in the abstract of line 12 and line 16 according to your suggestion. The “yields of” has been corrected to “yield of” in line 13. “planer” has been corrected to “planar” in line 23.

Question 10: Page 1, lines 36-41: This sentence is too long. Please, shorten it.

Answer: Thank you very much for your suggestions! We shorten the sentence in lines 36-41 as “Mangrove-derived microorganisms in intertidal microenvironments are tolerant to numerous stresses, including highly saline or brackish water, high solar irradiation, and tidal gradients. The special ecological niche induced certain buds being evolved to encode unique biosynthetic genes which have the potential to generate unique bioactive metabolites [1–2], which increasingly attracted the attention of both pharmaceutical and natural product chemists [3–4].” according to your suggestion.

Question 11: Page 2, line 36: “…1,300 new compounds” and page 13, line 442: “1,500 Da”

Page 2, line 63: “There are several insights…”

Page 2, line 66: “…phenols…”

Page 5, line 159; “resonations” should be “resonances”

Page 6, line 198: “…two nodes are numbered…”

Page 7, line 234: “…as well as comparison…”

Answer: Thank you! The “1300” has been corrected to “1,300” and the “1500” has been corrected to “1,500” in lines 52, 409-410, 440.

The extra a has been deleted in line 64.

The “phenol” has been corrected to “phenols” in line 67.

The “resonations” has been corrected to “ resonances” in line 161.

The “ modes” has been corrected to “ nodes” in line 201.

“comparation” has been revised to “comparison” in Page 7, line 234.

Question 12: Page 7, lines 233-234: Please, rephrase.

Answer: Thank you! The sentence in lines 233-234 was rephrased as “Besides, another node (singleton) at average m/z 599.1747 (compounds 1/2) that had no matches in the GNPS database,...”

Question 13: Page 9, Table 1: Please, check the character for compound dC, 9a

Answer: Thank you! The chemical shift of C-9a has been added in Page 9, Table 1.

Question 14: Page 13, line 433: Please, correct “LCMSoultion Ver3”

Answer: Thank you! The softerware was labled itself with “LCMSsoultion Ver3”. So we do not change the software name.

Question 15: Page 15, lines 517: Please, rewrite the sentence

 Answer: Thank you! We have changed “ All the cell data are presented as the mean standard deviation of the means (S.D.) and a one-way analysis of variance (ANOVA) was used to evaluate the statistical sig-nificance of differences between groups by Graphpad Prism.”

Question 16: References: The dois are missing

Answer: Thank you very much for your suggestions! The dois have been added in all the references.

We sincerely hope that this revised manuscript has addressed all your comments and suggestions. We appreciated for reviewers’ professional suggestions earnestly, and hope that the correction will meet with approval. Once again, thank you very much for your comments and suggestions.

Reviewer 2 Report

Comments and Suggestions for Authors

This manuscript describes the production of novel compounds through co-cultivation of two mangrove-derived fungal strains of the genus Phomopsis. Metabolomics analysis has detected many potential new compound, and three of them were actually purified and their structures were determined from spectral analysis. The provided spectral data are clear, and the planar structures and relative stereochemistry have been appropriately determined. The absolute stereochemistry is also considered correct, except for the points mentioned below. Judging from the database search, proposed structures are all new. The research is of a high standard as a search for new bioactive substances. It can be acceptable after the following corrections. 

1: Co-cultivation is widely used as a method to activate dormant biosynthetic genes, but many cases have issues with reproducibility. It should be demonstrated that the production of the compounds 1-3 is a reproducible result of co-cultivation by showing HPLC chromatograms in supplementary data, because many related compounds are produced by mono-culture conditions.

2: Related to the above comment, it is natural for each group to be separated in the PCA analysis of Figure S2. What is important is which compounds characterize each group. Did compounds 1-3 feature the co-cultivation group? A Loading Plot should also be shown.

3: The vertical axis of Figure 3 is incorrect.

4: For compound 3 in Figure 7, the calculated ECD spectra differ between compound 3 and its enantiomer. Also, there are four possible absolute configurations for compound 3. Each should be shown.

5: The text is overall too lengthy. Especially the sections before focusing on the compounds 1-3 should be more concise.

Author Response

Response to Reviewer :

Comments and Suggestions for Authors

This manuscript describes the production of novel compounds through co-cultivation of two mangrove-derived fungal strains of the genus Phomopsis. Metabolomics analysis has detected many potential new compound, and three of them were actually purified and their structures were determined from spectral analysis. The provided spectral data are clear, and the planar structures and relative stereochemistry have been appropriately determined. The absolute stereochemistry is also considered correct, except for the points mentioned below. Judging from the database search, proposed structures are all new. The research is of a high standard as a search for new bioactive substances. It can be acceptable after the following corrections.

Answer: Thank you very much for your positive comment and constructive suggestions! Please find the following detailed responses to your comments and suggestions. The modification of the manuscript are marked with a yellow background.

Question 1: Co-cultivation is widely used as a method to activate dormant biosynthetic genes, but many cases have issues with reproducibility. It should be demonstrated that the production of the compounds 1-3 is a reproducible result of co-cultivation by showing HPLC chromatograms in supplementary data, because many related compounds are produced by mono-culture conditions.

Answer: Thank you very much for your suggestions! At the beginning we carried out the HPLC chromatogram with UV detection (HPLC-UV) of the crude EtOAc extract of a 30-day solid rice medium whole co-culture of DHS-48 and DHS-11. The results showed significantly suppressed peaks and newly induced compounds 1-6 (including the new compounds 1-3) as shown in Figure 3. To ensure for the reproducible result of co-cultivation, we conducted five replicate groups of co-culture and its monoculture extracts for HPLC, PCA, 1HNMR, and MN analysis. Meanwhile, we added score plots (UPLC-MS data replicated five times) besides the loading plot in Figure S2A and S2B of supp. Info. The results all demonstrated that compounds 1-3 were produced only by co-culture and reproducible result of co-cultivation.

Question 2: Related to the above comment, it is natural for each group to be separated in the PCA analysis of Figure S2. What is important is which compounds characterize each group. Did compounds 1-3 feature the co-cultivation group? A Loading Plot should also be shown.

Answer: Thank you very much for your important suggestion!We selected the score plots to display the PCA analysis results in order to integrate the HNMR and HPLC results and investigate whether the two fungal strains' metabolic profiles changed following co-cultivation. By observing the score plots of co-culture and its monocultures, we found that the five replicate groups of DHS-48 monoculture were closely integrated, the five replicate groups of DHS-11 monoculture were closely integrated, and the five replicate groups after their co-culture were integrated, meanwhile the three types of biological samples were separated from each other obviously, indicating that their chemical compositions were different. We have added the loading plot besides score plots in Figure S2A and S2B of supp. Info.. The variable features responsible for discriminating these three groups are shown in the loading plots (Figure S2B). Compounds 1-3 are significant outliers in the loading diagram, indicating that they are the significant features responsible for distinguishing these three groups of variables. The UPLC-MS data were labled out in Figure S2B annotation according to your suggestion.

Question 3: The vertical axis of Figure 3 is incorrect.

Answer: Thank you! The vertical axis of Figure 3 has been corrected to “mAU”.

Question 4: For compound 3 in Figure 7, the calculated ECD spectra differ between compound 3 and its enantiomer. Also, there are four possible absolute configurations for compound 3. Each should be shown.

Answer: Thank you very much for your suggestions! We agreed that here are four possible absolute configurations for compound 3. As indicated by the comparison of 1H and 13C NMR data of compound 2 with compound 3, we found that the configuration of the left side of compound 3 should be 5’S, 6’S and 10a’R. The minor difference of the calculated and measured CD spectra and CD calculation of model compounds might due to the axial chirality along the biaryl axis dominates the CD spectra of the dimeric xanthone, which is expected to allow interconversion of the P and M helicity forms at room temperature as indicated by the literature “Elsässer B, Krohn K, Flörke U, et al. X-ray structure determination, absolute configuration and biological activity of phomoxanthone A. Eur. J. Org. Chem. 2005, 4563–4570”.

Question 5: The text is overall too lengthy. Especially the sections before focusing on the compounds 1-3 should be more concise.

Answer: Thank you very much for your suggestions! Since this work aimed at demonstrate the strategy combining co-cultivation with a metabolomics-guided strategy (integrated HPLC chromatogram, 1HNMR spectroscopy, UPLC-MS-PCA, and molecular networking) was a useful discovery tool for quick discovery of target bioactive compounds, the workflow and corresponding metabolomics results are all presented to make sure the conclusion were undoubted.

We sincerely hope that this revised manuscript has addressed all your comments and suggestions. We appreciated for reviewers’ professional suggestions earnestly, and hope that the correction will meet with approval. Once again, thank you very much for your comments and suggestions.

Reviewer 3 Report

Comments and Suggestions for Authors

In this manuscript, authors performed metabolomics-guided study of co-cultures of mangrove endophytic fungi Phomopsis asparagi DHS-48 and Phomopsis sp. DHS-11, leading to the discovery of nine dimeric xanthones including three new ones. Moreover, these compounds exhibited interesting cytotoxic and immunosuppressive activities. These findings were important, which was worth publishing in this journal.

However, revisions were required based on the following issues.

1.     Please check the MS and NMR data carefully. For instance, the HRESIMS data (m/z 599.1755 [M + H]+) reported in the manuscript was not consistent with the one (m/z 597.1613) shown in Figure S12. When describing the NMR data, the splitting pattern for δH 1.69 (d, J = 13.8 Hz) in Line 253 and the coupling constant for δH 1.95 (dd, J = 17.7, 4.2 Hz) in Line 254 did not match those displayed in Table 1. And don not forget to add ‘Hz’ after the coupling constant such as ‘δH 1.12, (d, J = 6.8)’ in Line 252.

2.     Although the relative configurations of the chiral centers in each mono xanthone unit of three new compounds were elucidated, the relationship for the configurations for these two units were not established before ECD calculations. For instance, what was the relationship for C-5′ and C-5 in compound 1? 5R*,5′R* or 5R*,5′S*? Please give an explanation.

3.     What initiative configurations used for ECD calculations for three new compounds? For instance, 5R,6R,8R,8aS,10aR,5’R,6’R,10a’S for compound 1? These information should be mentioned in the manuscript as well as Figure 7. Moreover, please provide the structures of the conformers mentioned in Supplementary Materials.

4.     It is better to give a preliminary analysis of the structure-activity relationships for these compounds in the manuscript.

Other revisions:

1.     ‘phomoxanthone L-N (13)’ → ‘phomoxanthones L-N (13)’

2.     Line 66: phenol’ → ‘phenols

3.     Other typo and grammar errors were also observed, such as the sentences ‘…isolation of new chemistry from complex mixtures…’(Line 88) and ‘…significant suppressed the proliferation…’(Line 528), Italic fonts for the species names ‘Phomopsis asparagi’, ‘Phomopsis sp.’(Page 3), ‘Rhizophora mangle’(Page 12) and the term ‘m/z’(Page 6), superscript and subscript fonts for ‘[α]20D’(Page 14).

4.     When downloading the references for review, some wrong information was found, such as the pages of [10], [22] and [63].

Comments on the Quality of English Language

There were a few grammar or typo errors. Some of them were given in the comments to the authors.

Author Response

Response to Reviewer :

Comments and Suggestions for Authors

In this manuscript, authors performed metabolomics-guided study of co-cultures of mangrove endophytic fungi Phomopsis asparagi DHS-48 and Phomopsis sp. DHS-11, leading to the discovery of nine dimeric xanthones including three new ones. Moreover, these compounds exhibited interesting cytotoxic and immunosuppressive activities. These findings were important, which was worth publishing in this journal.

Answer: Thank you very much for your positive comment and constructive suggestions! Please find the following detailed responses to your comments and suggestions. The modification of the manuscript are marked with a yellow background.

Question 1: Please check the MS and NMR data carefully. For instance, the HRESIMS data (m/z 599.1755 [M + H]+) reported in the manuscript was not consistent with the one (m/z 597.1613) shown in Figure S12. When describing the NMR data, the splitting pattern for δH 1.69 (d, J = 13.8 Hz) in Line 253 and the coupling constant for δH 1.95 (dd, J = 17.7, 4.2 Hz) in Line 254 did not match those displayed in Table 1. And don not forget to add ‘Hz’ after the coupling constant such as ‘δH 1.12, (d, J = 6.8)’ in Line 252.

Answer: Thank you very much for your constructive suggestions! We have revised the MS and NMR data for the manuscript and supplementary materials for all HRESIMS data using the positive mode and keep in consistant of the NMR data in the text with Table 1. The modifications have been highlighted, such as line 228, Fig. S3,4, lines 255-292.

Question 2: Although the relative configurations of the chiral centers in each mono xanthone unit of three new compounds were elucidated, the relationship for the configurations for these two units were not established before ECD calculations. For instance, what was the relationship for C-5′ and C-5 in compound 1? 5R*,5′R* or 5R*,5′S*? Please give an explanation.

Answer: Thank you very much for your comments! We tried to figure out the relative configuration of the two xanthone subunits before the ECD calculation mainly by the comparison of the 1H and 13C NMR with previously reported dimeric xantones. Take compound 1 as example, the configuration of the right subunit was compared with such as phomoxanthone D (compound 4), and the configuration of the left subunit was compared with phomopsis-H76 A(compound 5) and diaporthochromone B  (compound 6), which all co-isolated from our study. The chemical shift and coupling constants of the right side mono-subunit of compound 1 were very similar as elucidated the same as that of phomoxanthone D (compound 4). The chemical shift and coupling constants of the left side mono-subunit of compound 1 were very similar as that reported phomopsis-H76 A(compound 5). Meanwhile, the chemical shift and coupling constants of the left side mono-subunit of compound 2 were very similar as that reported right side mono-subunit of diaporthochromone B  (compound 6). Besides, since compound 1 was isolated together with phomoxanthone D (compound 4), phomopsis-H76 A(compound 5) and diaporthochromone B  (compound 6) in our study, it was expected the relationship for C-5′ and C-5 in compound 1 was assigned as 5R*,5′R* rather than 5R*,5′S* on the basis of biogenetic considerations.

Question 3: What initiative configurations used for ECD calculations for three new compounds? For instance, 5R,6R,8R,8aS,10aR,5’R,6’R,10a’S for compound 1? These information should be mentioned in the manuscript as well as Figure 7. Moreover, please provide the structures of the conformers mentioned in Supplementary Materials.

Answer: Thank you very much for your comments! The initiative configurations used for ECD calculations for three new compounds are compound1 (5R, 6R, 8R, 8aS, 10aR, 5’R, 6’R, 10a’S), compound 2(5R, 6R, 8R, 8aS, 10aR, 5’S, 6’S, 10a’R) and compound 3 (5S, 6R, 8R, 8aS, 10aR, 5’S, 6’S, 10a’R). According to your suggestion, we added the initiative configurations in the text as “Since compound 1 was co-isolated with 4-6 in our study, it was expected also to contain monomer with 5R, 6R, 8R, 8aS, 10aR absolute configuration on the basis of biogenetic considerations. Accordingly, four diastereomeric starting structures with (5R, 6R, 8R, 8aS, 10aR, 5’R, 6’R, 10a’S; 5S, 6S, 8S, 8aR, 10aS, 5’S, 6’S, 10a’R; 5R, 6R, 8R, 8aS, 10aR, 5’S, 6’S, 10a’R; 5S, 6S, 8S, 8aR, 10aS, 5’R, 6’R, 10a’S ) configuration of 1 and its diastereomer 2 were subjected to calculate the electronic circular dichroism (ECD) spectrum at the B3LYP/6-31+g (d, p) level to compare with the experimental circular dichroism (CD) spectrum (Figure 7).” and also in Figure 7 as following.

Figure 7. Experimental and calculated electronic circular dichroism (ECD) spectra of 1-3.

The structures of the conformers mentioned in Supplementary Materials in Table S2, S4 and S6 as following:

Table S2. Gibbs free energiesa and equilibrium populationsb of low-energy conformers of phomoxanthone L (1)

Conformers

In MeOH

Ga

P (%)b

1-1

-1342070.06533743

59.01

1-2

-1342069.80241074

37.85

1-3

-1342068.22798815

2.65

1-4

-1342067.2271097

0.49

aB3LYP/6-31G(d,p), in kcal/mol. bFrom G values at 298.15K.

Table S4. Gibbs free energiesa and equilibrium populationsb of low-energy conformers of phomoxanthone M (2)

Conformers

In MeOH

Ga

P (%)b

2-1

-1342069.61980533

29.44

2-2

-1342069.89026214

46.49

2-3

-1342067.9173707

1.66

2-4

-1342069.45853526

22.42

aB3LYP/6-31G(d,p), in kcal/mol. bFrom G values at 298.15K.

Table S6. Gibbs free energiesa and equilibrium populationsb of low-energy conformers of phomoxanthone N (3)

Conformers of 3

In MeOH

G a

P (%) b

3-1

-1342811.87691144

97.88

3-2

-1342809.50304111

1.77

3-3

-1342808.5423233

0.35

aG, B3LYP/ 6-31g (d, p), in kcal/mol. b Boltzmann-population.

Question 4: It is better to give a preliminary analysis of the structure-activity relationships for these compounds in the manuscript.

Answer: Thank you very much for your suggestions! We added some preliminary analysis of the structure-activity relationships for these compounds in “2.2. Biological Evaluation of Isolated Compounds” as following: “The results indicated that the tetrahydroxanthone moieties and hydroxyl groups attached at C-12 and C-12’ were the key functional architectures contributed to the cancer cells proliferation inhibitory effect.”

Question 5: The text is overall too lengthy. Especially the sections before focusing on the compounds 1-3 should be more concise.

Answer: Thank you very much for your comments! Since this work aimed at demonstrate the strategy combining co-cultivation with a metabolomics-guided strategy (integrated HPLC chromatogram, 1HNMR spectroscopy, UPLC-MS-PCA, and molecular networking) was a useful discovery tool for quick discovery of target bioactive compounds, the workflow and corresponding metabolomics results are all presented to make sure the conclusion were undoubted.

Question 6: Other revisions:

  1. ‘phomoxanthone L-N (1–3)’ → ‘phomoxanthones L-N (1–3)’

Answer: Thank you very much for your comments and reminding. The “ phomoxanthone L-N (1–3)” has been corrected to “ phomoxanthones L-N (1–3)” in lines 22,107, 525.

  1. Line 66: ‘phenol’ → ‘phenols’

Answer: Thank you very much for your comments and reminding. The “phenol” has been corrected to “phenols” in line 67.

  1. Other typo and grammar errors were also observed, such as the sentences ‘…isolation of new chemistry from complex mixtures…’(Line 88) and ‘…significant suppressed the proliferation…’(Line 528), Italic fonts for the species names ‘Phomopsis asparagi’, ‘Phomopsis’(Page 3), ‘Rhizophora mangle’(Page 12) and the term ‘m/z’(Page 6), superscript and subscript fonts for ‘[α]20D’(Page 14).

Answer: Thank you very much for your comments and reminding. ‘…isolation of new chemistry from complex mixtures…’(Line 88) has been revised as “target rapidly isolation of novel natural product from complex mixtures in co-cultivation”.

“Meanwhile, compounds 7-9 significant suppressed the proliferation of human liver cells HepG-2. Our study highlighted that combining co-cultivation with metabolomics-guided strategy as a discovery tool will be implemented as a systematic strategy for quick discovery of target bioactive compounds. ” has been corrected to “Meanwhile, compounds 7-9 significantly suppressed the proliferation of human liver cells HepG-2. Our study highlighted that combining co-cultivation with a metabolomics-guided strategy as a discovery tool will be implemented as a systematic strategy for quick discovery of target bioactive compounds. ”

Italic fonts for the species names ‘Phomopsis asparagi’, ‘Phomopsis sp.’(Page 3), ‘Rhizophora mangle’(Page 12) and the term ‘m/z’(Page 6) have been modified.

Superscript and subscript fonts for ‘[α]20D’(Page 14) has been modified in lines 472-478.

  1. When downloading the references for review, some wrong information was found, such as the pages of [10], [22] and [63].

Answer: Thank you very much for your comments and reminding. The references pages have been corrected.

We sincerely hope that this revised manuscript has addressed all your comments and suggestions. We appreciated for reviewers’ professional suggestions earnestly, and hope that the correction will meet with approval. Once again, thank you very much for your comments and suggestions.

Round 2

Reviewer 1 Report

Comments and Suggestions for Authors

Dear Authors, 

Thank you for this revised version. Although many improvements have been made, the manuscript still needs major changes before it can be accepted by the journal Marine Drugs.

 I do not dispute that your study included metabolomics analysis of one of the analytical techniques used but is not for the guidance of the new dimeric xanthones that you identified. 

The problem is the term “metabolomics-guided” because the whole study was not metabolomics-guided, especially with the statistical analysis that was performed. If I understand your study correctly, each monoculture and co-culture was analyzed 5 times. Therefore, the PCA analysis tested the analytical repeatability of the UPLC-ESI-MS/MS technique and not the bio-variability.

The PCA algorithm is based on the detection of the variability of independent samples. This means that samples are obtained from different biological sources and analyzed independently (data acquisition and processing). The PCA model aims to separate the samples according to the variance observed. Here, the model was computed with 3 biological samples (2 monocultures and 1 co-culture) and the UPLC-ESI-MS/MS analysis was repeated 5 times for each sample. This means that the PCA model tested the technical repeatability. The model should be calculated with 5 independent cultures of the three samples, which can greatly reduce its reliability.

The term “metabolomics” needs to be associated with the correct keywords.

After reading your reply, I suggest that you correct the title of your manuscript and use “Computational methods guided…”, delete “metabolomics-guided” throughout the manuscript and delete the supplementary figure S2, which provides no additional information beyond the HPLC profiles but confusing information, as well as the corresponding part in the main text.

In addition, the workflow should be modified accordingly and moved after line 175.  

Question 3: I suggest adding a sentence, similar to the one in the paper by Deniz Tasdemir et al. (Int. J. Mol. Sci. 2022, 23, 782), explaining why the three new dimeric xanthones are not located in the same cluster (cosine < 0.65). 

Question 4: I am surprised by your answer because SIRIUS, MolDiscovery and MolNetEnhancer for example are free tools available online. I encourage you to use them.  

Question 6: Please, improve the quality of Figure 4. 

Question 14: “LCMSsoultion Ver.3” should be “LCM solution Ver.3”.

Comments on the Quality of English Language

The quality of the English is good.

Author Response

Response to Reviewer :

Comments and Suggestions for Authors

Question 1: I do not dispute that your study included metabolomics analysis of one of the analytical techniques used but is not for the guidance of the new dimeric xanthones that you identified. 

The problem is the term “metabolomics-guided” because the whole study was not metabolomics-guided, especially with the statistical analysis that was performed. If I understand your study correctly, each monoculture and co-culture was analyzed 5 times. Therefore, the PCA analysis tested the analytical repeatability of the UPLC-ESI-MS/MS technique and not the bio-variability.

The PCA algorithm is based on the detection of the variability of independent samples. This means that samples are obtained from different biological sources and analyzed independently (data acquisition and processing). The PCA model aims to separate the samples according to the variance observed. Here, the model was computed with 3 biological samples (2 monocultures and 1 co-culture) and the UPLC-ESI-MS/MS analysis was repeated 5 times for each sample. This means that the PCA model tested the technical repeatability. The model should be calculated with 5 independent cultures of the three samples, which can greatly reduce its reliability.

The term “metabolomics” needs to be associated with the correct keywords.

After reading your reply, I suggest that you correct the title of your manuscript and use “Computational methods guided…”, delete “metabolomics-guided” throughout the manuscript and delete the supplementary figure S2, which provides no additional information beyond the HPLC profiles but confusing information, as well as the corresponding part in the main text.

Answer: Thank you very much for your comments and reminding. We are terribly sorry for our mistakes that we did not accurately describe the part of sample preparation for PCA analysis, “each monoculture and co-culture was analyzed 5 times” actually refers to we randomly selecting 5 bottles from 130 co-cultured fermentation mixes, extracting and preparing samples separately. Similarly, we also randomly selecting 5 bottles from 20 single cultured fermentation products to extract and prepare the corresponding crude extracts for PCA analysis. This can ensure the accuracy and independence of the results, rather than analyzing the same sample 5 times. We have made corresponding modifications in lines 181-183 and lines 392-412 and highlighted it.

lines 181-183:

All co-cultures and monocultures have five independent biological replicates, ensuring the reliability and independently of the results [55].

lines 392-412:

3.4. Sample Preparation of Phomopsis asparagi, Phomopsis sp., Co-culture and Large-Scale Fermentation and Extracts

The two fungi were independently cultivated on PDA at 28 °C for 14 days. After that the two fungi colonies were simultaneously inoculated into an autoclaved rice sold-substrate medium in Erlenmeyer flasks (130 × 1 L), each contained 100 g of rice, 100 mL of 0.3% saline water and fermented at 28 °C for 30 days. At the same time, Phomopsis asparagi and Phomopsis sp. were separately cultured under the same culture conditions in 20 Erlenmeyer flasks.

Following the fermentation process, a random selection of 5 bottles was made from the 130 co-cultured fermentation mixes, and similarly, 5 bottles were randomly chosen from the 20 bottles of Phomopsis asparagi and Phomopsis sp. monocultured fermentation mixes. The co-cultured and monocultured fermentation mixes that were selected were extracted three times with EtOAc and the filtrate was then distilled under reduced pressure to get the crude extracts. The above operations are to ensure that all co-cultures and monocultures have five independent biological replicates, ensuring the reliability and independently of the results. The dried extracts were re-dissolved in an appropriate amount of UPLC/MS grade methanol (MeOH) and pipetted into a pre-weighed 1.5mL-amber glass vial through a 13 mm syringe filter with a 0.22 mm PTFE membrane to prepare a 1mg.mL-1 solution for future use. The crude extracts were analyzed using HPLC, UPLC-MS/MS and 1H NMR. The remaining 125 bottles of the co-cultured fermentation mixes were similarly extracted three times with EtOAc to get 30 g of crude extract for subsequent isolation.

About the changing of the term “metabolomics-guided” into “Computational methods guided…”, our research thread is as Scheme 1 illustrated. Metabolomics is defined as a combination of multiparametric chemical analysis techniques and multivariate statistical analyses.

We first performed HPLC-UV (which do not included any computational method) indicated that showed unexpected and significantly suppressed peaks and newly induced compounds 1-6, which were absent in the monocultures. The common characteristic UV absorption maxima around 203, 225, 347 nm (UV data of compound 7) showed a coincidence to related literature values [54], indicating that 1-9 should be dimeric xantone derivatives. 1H NMR metabolic profile (Figure S1) (which also do not included any computational method)  of EtOAc extracts showed several significant enhanced hydrogen resonances at 6.5-7.7 ppm (aromatic protons), 3.5-4.5 ppm (alkoxy protons), and 2.5-2.75 ppm (methine/methylene protons) characterized for xantone derivatives. Principal component analysis (PCA) was conducted to compare the metabolic features and identify statistically significant differences, and we also illustrated the m/z of dimeric xantone derivatives, that’s why we put this Figure S2 in the supporting information.  In order to further investigate the metabolome and to obtain more information on the chemical diversity induced by co-culture, a UPLC-ESI-MS/MS based molecular networking was generated with the crude extracts of the fungal co-culture and the corresponding monocultures through the online Global Natural Products Social Molecular Networking (GNPS) platform. Overall, our aim of the work is surrounding the new metabolomics approach (integrated HPLC chromatogram, 1HNMR spectroscopy, UPLC-MS-PCA, and molecular networking) guided the targeted isolation of the induced metabolites. So we insisted of the using of word “metabolomics-guided” instead of “Computational methods guided…”. Thank you again for your comments and kind suggestions!

Question 2: In addition, the workflow should be modified accordingly and moved after line 175. 

Answer: Thank you very much for your reminding! We have added the workflow in Scheme 1 in line 116 as: “Scheme 1. Workflow of targeted isolation of new dimeric xanthones from co-cultures of mangrove endophytic fungi Phomopsis asparagi DHS-48 and Phomopsis sp. DHS-11 based on integrated metabolomics-guided discovery.”

Question 3:  I suggest adding a sentence, similar to the one in the paper by Deniz Tasdemir et al. (Int. J. Mol. Sci. 2022, 23, 782), explaining why the three new dimeric xanthones are not located in the same cluster (cosine < 0.65).

Answer: Thank you very much for your suggestions!  We added the aforementioned reference to explain why the three new dimeric xanthones are not located in the same cluster in lines 250-252 according to your suggestions.

Question 4: I am surprised by your answer because SIRIUS, MolDiscovery and MolNetEnhancer for example are free tools available online. I encourage you to use them. 

 Answer: Thank you very much for your suggestions! Since this work is our first GNPS analysis, we have already included several free database for annotation. We may try to use these database in the future works.

Question 6: Please, improve the quality of Figure 4.

 Answer: Thank you very much for your reminders. We have already improved the graphic resolution of Figure 4.

Question 14: “LCMSsoultion Ver.3” should be “LCM solution Ver.3”.

Answer: Thank you very much for your reminders. We have changed “LCMSsoultion Ver.3” to “LCM solution Ver.3” according to your suggestion in line 453.

Round 3

Reviewer 1 Report

Comments and Suggestions for Authors

The manuscript can now be accepted for publication in Marine Drugs.

Comments on the Quality of English Language

The quality of the English is good.